# Computational gene expression analysis reveals distinct molecular subgroups of T-cell prolymphocytic leukemia

Nathan Mikhaylenko[1], Linus Wahnschaffe[2,3,4], Marco Herling[2,3,4,5], Ingo Roeder[1,6,7,8,9], Michael Seifert ![ORCID][1,6,7,8,9] *

1 Institute for Medical Informatics and Biometry (IMB), Carl Gustav Carus Faculty of Medicine, Technische Universität Dresden, Dresden, Germany, 2 Department I of Internal Medicine, Center for Integrated Oncology (CIO), Aachen-Bonn-Cologne-Duesseldorf, University of Cologne, Cologne, Germany, 3 Excellence Cluster for Cellular Stress Response and Aging-Associated Diseases (CECAD), University of Cologne, Cologne, Germany, 4 Center for Molecular Medicine Cologne (CMMC), University of Cologne, Cologne, Germany, 5 Department of Hematology and Cellular Therapy, University of Leipzig, Leipzig, Germany, 6 National Center for Tumor Diseases (NCT), Dresden, Germany, 7 German Cancer Research Center (DKFZ), Heidelberg, Germany, 8 Faculty of Medicine and University Hospital Carl Gustav Carus, Technische Universität Dresden, Dresden, Germany, 9 Helmholtz-Zentrum Dresden - Rossendorf (HZDR), Dresden, Germany

* michael.seifert@tu-dresden.de

**Data Availability Statement:** All relevant data are within the manuscript and its Supporting information files. All data analyses were performed using public R packages. The sources of the

## Abstract

T-cell prolymphocytic leukemia (T-PLL) is a rare blood cancer with poor prognosis. Overexpression of the proto-oncogene *TCL1A* and missense mutations of the tumor suppressor *ATM* are putative main drivers of T-PLL development, but so far only little is known about the existence of T-PLL gene expression subtypes. We performed an in-depth computational reanalysis of 68 gene expression profiles of one of the largest currently existing T-PLL patient cohorts. Hierarchical clustering combined with bootstrapping revealed three robust T-PLL gene expression subgroups. Additional comparative analyses revealed similarities and differences of these subgroups at the level of individual genes, signaling and metabolic pathways, and associated gene regulatory networks. Differences were mainly reflected at the transcriptomic level, whereas gene copy number profiles of the three subgroups were much more similar to each other, except for few characteristic differences like duplications of parts of the chromosomes 7, 8, 14, and 22. At the network level, most of the 41 predicted potential major regulators showed subgroup-specific expression levels that differed at least in comparison to one other subgroup. Functional annotations suggest that these regulators contribute to differences between the subgroups by altering processes like immune responses, angiogenesis, cellular respiration, cell proliferation, apoptosis, or migration. Most of these regulators are known from other cancers and several of them have been reported in relation to leukemia (e.g. *AHSP, CXCL8, CXCR2, ELANE, FFAR2, G0S2, GIMAP2, IL1RN, LCN2, MBTD1, PPP1R15A*). The existence of the three revealed T-PLL subgroups was further validated by a classification of T-PLL patients from two other smaller cohorts. Overall, our study contributes to an improved stratification of T-PLL and the observed subgroup-specific molecular characteristics could help to develop urgently needed targeted treatment strategies.

corresponding R codes and details to settings are given in the materials and methods section. Utilized R scripts and corresponding data sets are publicly available from Zenodo at https://doi.org/10.5281/zenodo.6586472.

**Funding:** This work was done within the Transcan-2 ERA-NET consortium 'ERANET-PLL' funded by the EU Horizon 2020 program (grant numbers: 01KT1906A/B). We also acknowledge support by the German Research Foundation and the Open Access Publication Funds of the SLUB/TU Dresden to cover the article processing charge. The funders had no role in study design, data collection and analysis, decision to publish, or preparation of the manuscript.

**Competing interests:** The authors have declared that no competing interests exist.

## Introduction

T-cell prolymphocytic leukemia (T-PLL) is a rare but highly malignant mature T-cell leukemia with aggressive clinical course and high mortality rates [1–5]. T-PLL was first described by [6] almost 50 years ago. T-PLL represents less than 2% of all mature lymphocytic leukemias [7], but it is still the most frequent mature T-cell leukemia in Western countries with an incidence of approximately two cases per one million people per year [8]. T-PLL mainly affects elderly people with a median age of 65 years at diagnosis [4, 7]. The median overall survival of T-PLL patients from diagnosis is less than three years [3, 9]. The widely considered first line monoclonal antibody therapy with alemtuzumab, which is effective in more than 80% of T-PLL patients, is typically followed by a relapse of nearly all patients at a median time of one year [3]. Due to the increased age of T-PLL patients, only about 30–50% of patients are eligible for an allogeneic hematopoietic stem cell transplantation, which represents the only curative option that currently exists [3]. Therefore, strong efforts are necessary to identify novel compounds to improve patient outcomes [3, 9–13].

Nowadays, T-PLL is diagnosed based on uniform diagnosis criteria that consider the histological presence of clonal prolymphocytic T-cells, the presence of complex chromosomal aberrations (e.g. inversions or translocations of chromosome 14), and a typical clinical representation to distinguish T-PLL from other T-cell leukemias [14, 15]. T-PLL patients do often present with exponentially rising lymphocyte counts in peripheral blood and bone marrow infiltration already at diagnosis, reflecting the uncontrolled proliferation of mature prolymphocytic tumor T-cells [2, 4]. Due to the aggressiveness of the disease, clinical features such as splenomegaly and hematological disorders (e.g. anemia, thrombocytopenia) are frequently observed symptoms [4]. In some rare cases, patients can also show a primarily inactive disease state, which eventually progresses into an active T-PLL over time [14, 16].

Increasing evidences suggest that chromosomal aberrations are a main driver of T-PLL development. Especially the *TCL1* gene family, the *ATM* gene, and DNA copy number alterations affecting chromosome 8 have been identified as pathogenic or prognostic factors of T-PLL [4, 5]. In most cases, complex T-PLL karyotypes show an overexpression of the proto-oncogene *TCL1A* at chromosome 14q32.1 due to an inversion (inv(14)(q11;q32)) or a translocation (t(14;14)(q11;q32)) [4, 5, 17]. This activation of *TCL1A* is frequently observed in combination with the inactivation of the tumor suppressor *ATM* at chromosome 11q22.3 by deletions and/or missense mutations [17–19]. This combination of a *TCL1A* overexpression with a damaging *ATM* aberration is thought to represent a potential key event that initiates T-PLL development by contributing to impaired DNA damage repair and abrogated p53-mediated cell death [17]. However, also T-PLL cases that lack the activation of *TCL1A* have been reported [7, 17]. Thus, the landscape of molecular lesions that contribute to T-PLL pathogenesis is complex and still not completely understood. Other recurrently observed molecular alterations such as translocations affecting *MTCP1* (a homolog of *TCL1A*) or haplo-insufficiency of *CDKN1B* can also influence cell cycle, apoptosis and DNA repair and thereby contribute to T-PLL development [4]. In addition, observed recurrent mutations of epigenetic regulators (*EZH2*, *TET2*, *BCOR*), of the DNA damage regulator *CHEK2*, and of Jak-Stat signaling genes (*IL2RG*, *JAK1*, *JAK3*, *STAT5B*) further increase the complexity of pathomechanisms that can contribute to T-PLL development [20–22].

Despite these advances, a more detailed understanding of the complex molecular alterations in T-PLL is required to further improve the stratification of patients. So far, only little is known about the existence of T-PLL gene expression subtypes. A main reason for this is the rarity of T-PLL that also restricts the number of available molecular profiles. Therefore, studies that analyzed the transcriptional landscape of T-PLL have mainly focused on the comparison

of T-PLL to normal controls. For example, differentially expressed genes involved in lympho-magenesis, cell cycle regulation, apoptosis and DNA repair were identified by [23] comparing five T-PLL samples to eight normal blood controls. Overall, *TCL1A* has shown the strongest upregulation in different studies [10, 17, 24]. Also the proto-oncogene *MYC*, the miRNA-processing regulator *AGO2*, and two other *TCL1* family members, *TCL1B* and *MTCP1*, have shown increased expression in comparison to normal controls [10, 17]. Further, a recent study by [25] did not reveal clear T-PLL subgroups based on gene expression profiling, but highlighted an important potential role of oncogenic miRNAs in T-PLL. Similarly, altered miRNA regulatory networks in T-PLL and their impacts on DNA damage response and cell survival have been revealed in [26].

With the recent availability of molecular data of a larger T-PLL patient cohort [17], a systematic computational search for robust T-PLL gene expression subgroups is now technically feasible. The identification of such T-PLL subgroups can help to improve patient stratification and may contribute to the development of urgently needed targeted treatment strategies for individual T-PLL patients.

Here, we performed a computational analysis of the T-PLL data set from [17] with the goal to identify and characterize potentially existing T-PLL subgroups. Three robust T-PLL gene expression subgroups were identified by hierarchical clustering in combination with boot-strapping. Similarities and differences between these subgroups were further determined at the level of individual genes, signaling and metabolic pathways, gene copy number alterations, and gene regulatory networks. The existence of these T-PLL subgroups was further supported by analyses of patients from two other T-PLL cohorts. Our analysis contributes to a better molecular stratification of T-PLL patients and may provide important information for the development of targeted treatment strategies.

## Results

### Genome-wide gene expression analysis reveals three distinct T-PLL subgroups

Hierarchical clustering of genome-wide gene expression profiles of 68 T-PLL samples and 10 healthy control samples was performed to characterize differences between T-PLL tumor T-cells and normal CD3$^+$ pan T-cells (Fig 1). This resulted in a grouping of all samples into four major subclusters. One of these subcluster exclusively represented all normal control samples (Fig 1, red subcluster, n = 10), whereas the other three subclusters represented the T-PLL samples according to their specific gene expression patterns. Interestingly, the first T-PLL subgroup (Fig 1, SG1: blue subcluster, n = 18) was located in the same subtree like the normal control group. The other two T-PLL subgroups SG2 and SG3 (Fig 1, SG2: green subcluster, n = 11; SG3: orange subcluster, n = 39) were located in a separate subtree. Thus, SG2 and SG3 were more similar to each other in comparison to SG1.

The stability of the obtained subgroups was first analyzed by bootstrapping of genes [27], which repeats the hierarchical clustering of all patients based on randomly chosen subsets of genes (S1A Fig). The control group and the two T-PLL subgroups SG2 and SG3 showed good moderate stabilities (AU values about 80%), whereas the T-PLL subgroup SG1 was less stable (AU value 62%). Overall, the stability of the subclusters in the dendrogram further increased towards the leaves representing the individual samples. In addition, also the stability of patient assignments to the three T-PLL subgroups was analyzed. Even for random removals of ten patients, the hierarchical clustering remained very stable with a median correctness of 95% for the reconstruction of the initially observed subgroups (S2A Fig).

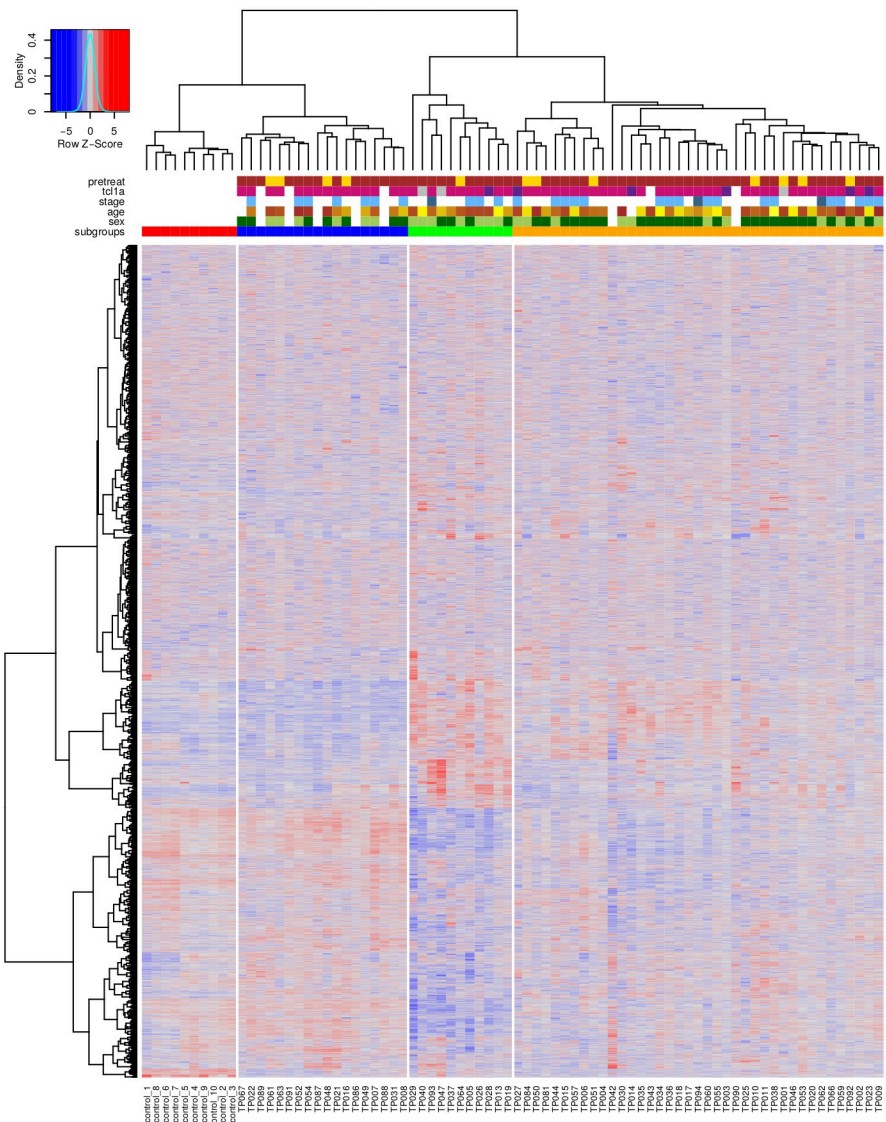

**Fig 1. Genome-wide clustering of T-PLL and normal control expression profiles.** Heatmap representing z-score-scaled expression measurements of 17,970 genes of each sample highlighting reduced (blue), unchanged (grey), and increased (red) expression of each gene in a specific sample in comparison to all other samples. T-PLL and normal control samples (columns) were clustered based on the similarity of their expression profiles and their corresponding gene-specific expression values are visualized (rows). The column dendrogram above the heatmap represents the clustering of individual samples defining four major subgroups: healthy control subcluster (red) and three T-PLL subclusters (blue: SG1, green: SG2, orange: SG3) as shown in the annotation column 'subgroups' below the dendrogram. Additional annotation columns contain further patient-specific meta-information: pretreatment before sample acquisition (brown: no, yellow: yes), *TCL1A* gene activity status based on immunophenotypic protein expression (pink: positive, violet: negative, grey: weak, white: NA), disease stage (light blue: active, dark blue: inactive, white: NA), age (range from light—32 years—to dark yellow—78 years—white: NA), and sex (light green: male, dark green: female, white: NA). Individual samples names are shown in the corresponding columns below the heatmap and the dendrogram left to the heatmap represents the clustering of genes.

Meta-information about treatment of patients before sample acquisition, sex, age, disease stage, and immunophenotypic TCL1A surface marker expression of patients of the three T-PLL subgroups have been mapped to the hierarchical clustering (Fig 1, annotation matrix below column dendrogram, S1 Table). The majority of T-PLL samples was from untreated

patients at diagnosis, but 12 of 68 samples were from patients in relapse after a previous treatment (median time of sample acquisition after end of treatment: 120.5 days). Each of the three revealed T-PLL subgroups contained at least one pretreated patient (SG1: 4 of 18, SG2: 1 of 11, SG3: 7 of 39). The proportion of pretreated patients did not significantly differ between the subgroups (Fisher's exact test: $p = 0.75$). Thus, pretreated patients did not strongly influence the identification and molecular composition of the revealed T-PLL subgroups. The sex distribution was balanced for SG1 and SG2 (with male to female ratios of 6:5 and 3:5) and strongly skewed for SG3 with a higher number of male patients (24:9), but not significantly different (Fisher's exact test: $p = 0.08$). The age was distributed equally and did not show any specific pattern that distinguished the three T-PLL subgroups. Further, almost all T-PLL patients for which a disease stage classification was available had an active disease, except for three patients that were annotated as inactive. In addition, almost all T-PLL patients were annotated to show an immunophenotypic expression of the TCL1A driver protein. In more detail, all patients of SG1 were TCL1A positive, whereas in few cases a weak or no immunophenotypic expression of TCL1A was reported for SG2 and SG3. T-PLL patients with a positive TCL1A protein expression status showed significantly greater *TCL1A* gene expression than patients with weak or no TCL1A protein expression (S3 Fig, Wilcoxon rank sum test: $p = 0.0004$ for positive vs. negative, $p = 0.01$ for positive vs. weak).

A Kaplan-Meier analysis was performed to analyze whether the patients of three revealed T-PLL expression subgroups differed in their survival. Overall, the survival of T-PLL patients from diagnosis did not strongly differ between the three subgroups (S4A Fig). However, females had a higher risk independent of the T-PLL subgroup ($p = 0.01$, S4B Fig), but an additional differential gene expression analysis of male and female patients did not show strong sex-specific expression differences. Therefore, other factors such as the immune system may potentially contribute to this observation [28, 29].

### Differential gene expression analysis of T-PLL subgroups

To identify genes that differ in their expression or that are commonly altered in the three revealed T-PLL subgroups, we determined differentially expressed genes for each subgroup in comparison to the normal control samples (Fig 2, S4 Table). Generally, more down- than up-regulated genes were found for each subgroup (Fig 2A). SG1 showed a much smaller number of significantly altered genes than SG2 or SG3 at the q-value cutoff of 5% (Fig 2B, left Venn diagram, SG1: 1,726, SG2: 4,271, SG3: 3,060). In addition, a separate analysis of down- and up-regulated genes showed a strong overlap of affected genes for SG2 and SG3, whereas SG1 shared much less commonly altered genes with both of them (Fig 2B, middle and right Venn diagram). Especially for the comparison of SG1 and SG2, a large proportion of genes that was down-regulated in one subgroup was up-regulated in the other subgroup and vice versa (Fig 2B). All these findings are in good accordance with the global gene expression patterns that distinguished the three revealed T-PLL subgroups (Fig 1).

The identified differentially expressed genes of the individual T-PLL subgroups were further analyzed for the enrichment of cancer-relevant gene annotation categories (S5A Fig). Similar annotation patterns were observed for all three subgroups with gene counts proportional to the number of altered genes. Especially the gene sets of essential genes and cancer census genes were significantly enriched in each subgroup. Many signaling pathway genes and transcription factors/co-factors were also altered in each subgroup and significantly enriched in SG2 and SG3. Still, the overlap of genes between the subgroups was relatively small ranging from 1.6% for tumor suppressors up to 6.7% for kinases (S5B Fig).

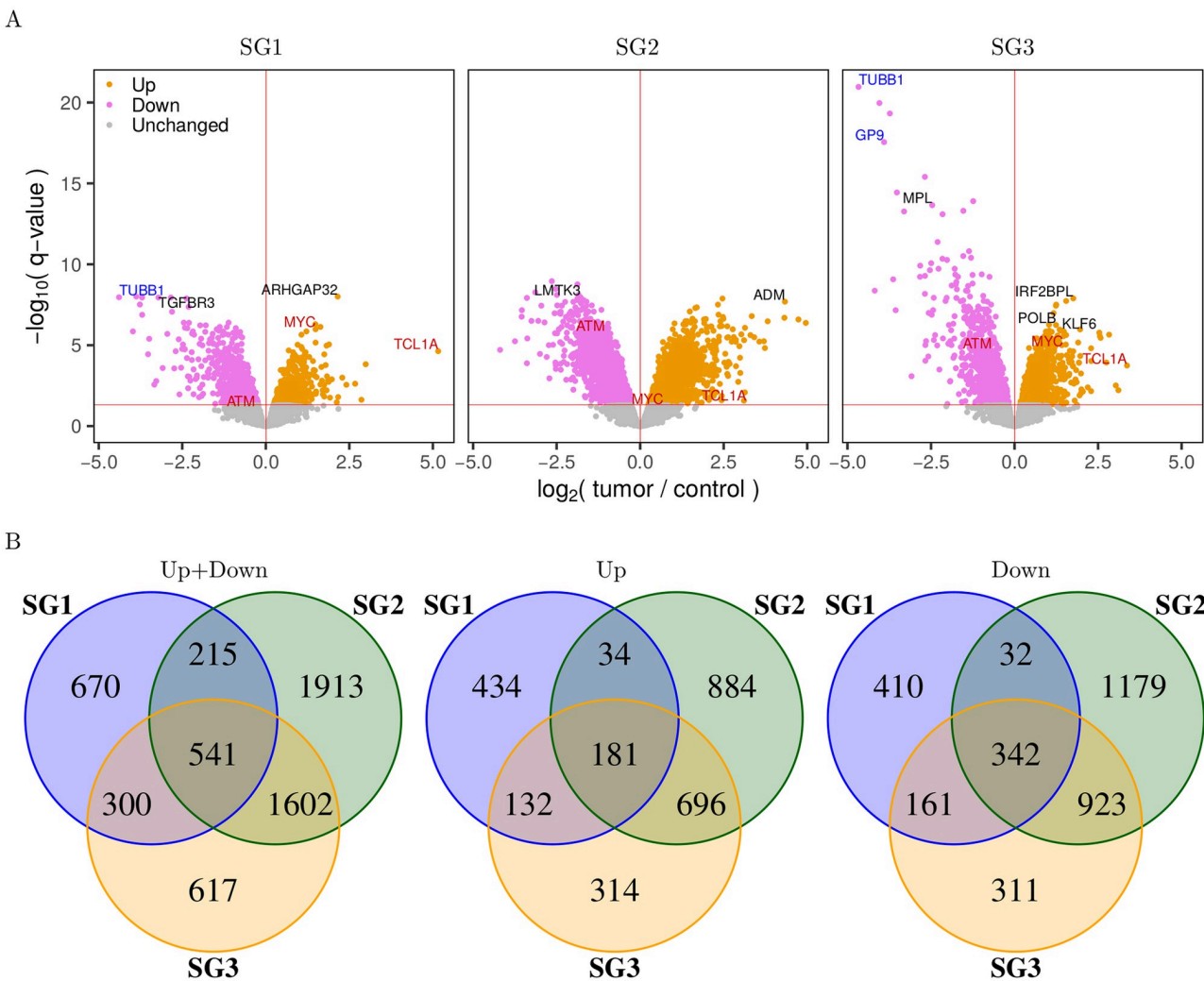

**Fig 2. Differential gene expression analysis of T-PLL subgroups.** A: Volcano plots of differential gene expression analysis of each T-PLL subgroup in comparison to the normal control group. The red horizontal dashed line marks the significance cutoff for the q-value cutoff 0.05. Genes above this line were considered as up-regulated (brown) or as down-regulated (lilac) in the T-PLL subgroup compared to the normal control group. Genes below this line were considered unchanged (gray). Some selected gene names are explicitly shown (red: known putative T-PLL drivers, blue: potential major regulators derived from our network, black: other known cancer-relevant genes with strong expression changes). B: Venn diagrams of differentially expressed genes at the q-value cutoff 0.05 showing similarities and differences between the three T-PLL subgroups. The left panel shows any significant gene, independent of the sign of fold change (both up- and down-regulated), the middle panel shows only up-regulated genes, and the right panel shows only down-regulated genes.

Next, we analyzed how well the identified 5,858 differentially expressed genes at the q-value cutoff of 5% were able to distinguish between the three T-PLL subgroups. Therefore, we reclustered all samples based on these genes and found that all T-PLL samples were again assigned to their previous subgroup, except TP067 which switched from SG1 to SG3 (S6 Fig). We also analyzed the stability of this clustering by bootstrapping of genes [27]. Compared to the previous stability analysis of the hierarchical clustering of all genes, the robustness was strongly increased for all subgroups (S1B Fig, AU values: 100% control, 88% SG1, 93% SG2, 91% SG3). Such an improvement was also observed for repeated hierarchical clusterings based on subsets of all patients, which now reached a median correctness of 97% for the reconstruction of the initially observed subgroups (S2B Fig). Thus, the identified differentially expressed genes cover

characteristic expression differences between the three T-PLL subgroups enabling to assign individual heterogeneous T-PLL expression profiles to their corresponding subgroup.

We also specifically analyzed the subgroup-specific expression behavior of the two putative main T-PLL driver genes *TCL1A* and *ATM* [17]. The proto-oncogene *TCL1A* was significantly up-regulated in all three T-PLL subgroups compared to the normal controls, but the degree of up-regulation was different between the subgroups. The *TCL1A* expression levels were on average greater in SG1 than in SG2 and SG3 (S7 Fig, S4 Table, average $\log_2$-fold change for T-PLL vs. control: 5.16 for SG1, 1.84 for SG2, 3.37 for SG3). Similarly, the tumor suppressor gene *ATM* was only significantly down-regulated in SG2 and SG3 but not in SG1 (S7 Fig, S4 Table, average $\log_2$-fold change for T-PLL vs. control: -0.32 for SG1, -1.06 for SG2, -0.69 for SG3). Thus, these subgroup-specific expression differences of both genes and their counteracting roles may have an impact on T-PLL subgroup development and progression.

## Similarities and differences of signaling and metabolic pathway alterations of T-PLL subgroups

The observed gene expression differences of the three T-PLL subgroups further motivated us to characterize similarities and differences between the subgroups at the level of cancer-relevant signaling and metabolic pathways. We therefore performed a pathway enrichment analysis for the down- and up-regulated genes of each subgroup (S4 Table, differentially expressed genes at q-value cutoff of 5%). In accordance with our differential gene expression analysis, more down- then up-regulated genes were observed for the individual signaling pathways (Fig 3A–3C). Focusing on significant enrichments, SG1 showed an enrichment of down-regulated genes for cytokine receptor interaction, apoptosis, and focal adhesion (Fig 3A, q ≤ 0.05). SG2 showed an enrichment of down-regulated genes for nucleotide excision repair and an enrichment of up-regulated genes for MAPK signaling and apoptosis (Fig 3B, q ≤ 0.05). SG3 showed an enrichment of up-regulated genes for nucleotide excision repair (Fig 3C, q ≤ 0.01). Generally, signaling pathway alteration profiles were globally very similar between the subgroups, but each subgroup had still its own characteristic signaling pathway enrichment profile.

Further considering the altered genes of individual signaling pathways, we observed that the number of overlapping genes between SG2 and SG3 was much greater than between SG1 and these two subgroups (S8A Fig). This is in accordance with the location of the subgroups in the hierarchical clustering (Fig 1). We illustrated this in Fig 4 for the Jak-Stat signaling pathway that plays an important role in T-cell related cancers [9, 22]. We focused on up-regulated Jak-Stat genes, because they could potentially be targeted by existing drugs. Four of the seven up-regulated genes in SG1 were unique for SG1, one was shared with SG3 (*MYC*), and two of these genes were shared with SG2 and SG3 (*CSF3R*, *IFNGR2*), whereas SG2 and SG3 had six other altered genes in common (*CREBBP*, *EP300*, *IL11*, *PIAS2*, *PIK3CA*, *RAF1*) apart from their own uniquely altered Jak-Stat signaling genes.

Interestingly, the metabolic pathway analysis showed a clearly distinct pattern for SG1 in comparison to SG2 and SG3, which both showed similar global metabolic alteration patterns (Fig 3D–3F). In addition, SG1 had more up- than down-regulated metabolic pathway genes, whereas SG2 and SG3 showed more down- than up-regulated genes. Consequently, SG2 and SG3 shared also more commonly altered metabolic pathway genes in comparison to SG1 at the level of individual pathways (S8B Fig). For example, none of the up-regulated glycolysis or citric acid cycle genes from SG1 was shared with SG2 or SG3. Further, SG2 and SG3 shared the significant enrichment of up-regulated genes of the pyruvate and purine metabolism, which was not observed for SG1 (Fig 3E and 3F, q ≤ 0.05). Moreover, the expression behavior of the oxidative phosphorylation pathway differed. SG1 showed clearly more up-regulated

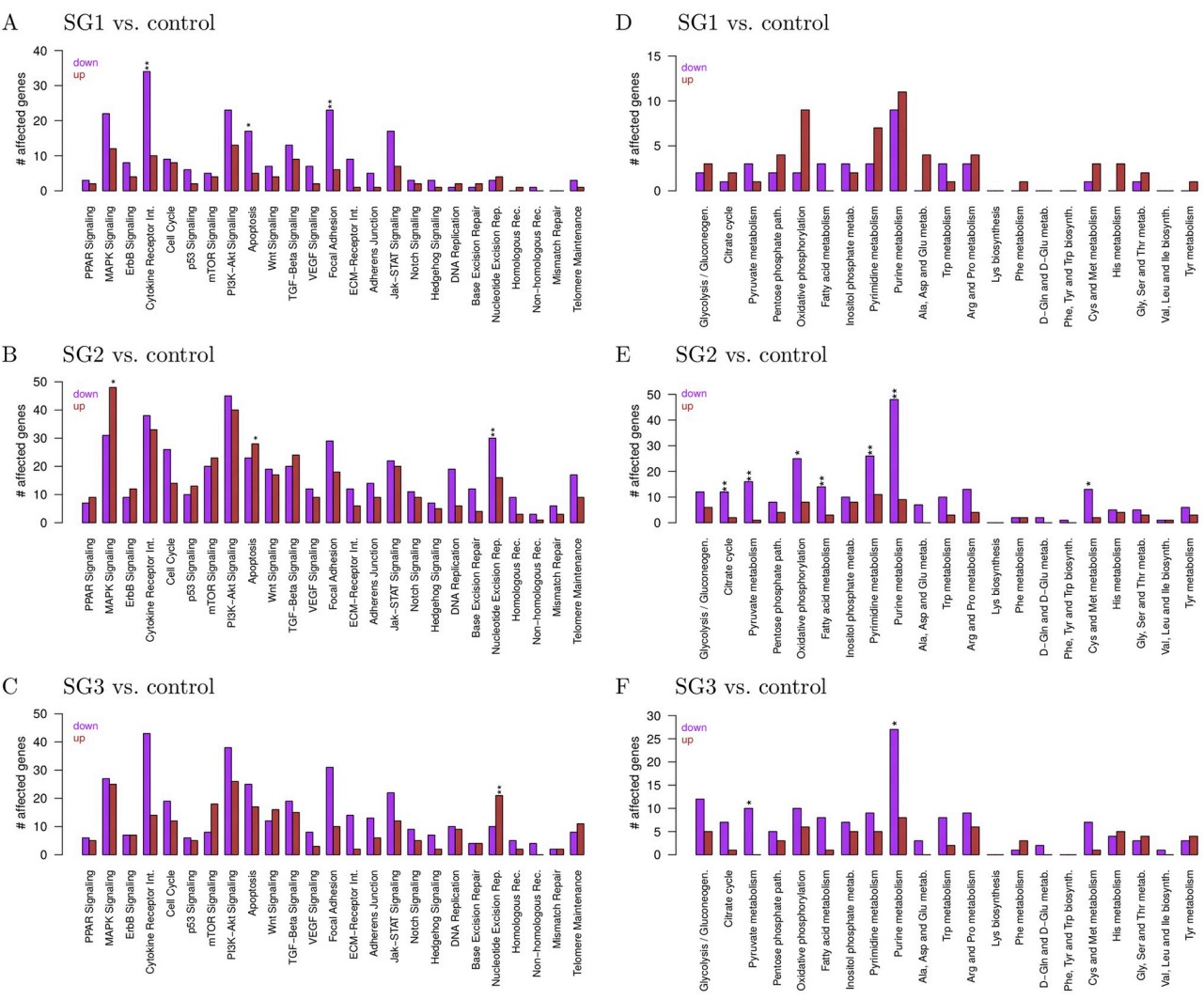

**Fig 3. Signaling and metabolic pathway analysis of differentially expressed genes of each T-PLL subgroup.** All down- (lilac) and up-regulated (brown) genes of a T-PLL subgroup in comparison to the normal control references were considered at the q-value cutoff of 5%. Enriched pathways are marked by asterisks (Fisher's exact test: '**': q ≤ 0.01, '*': q ≤ 0.05). A–C: Cancer signaling pathway analysis for subgroups SG1, SG2, and SG3. D–F: Metabolic pathway analysis for SG1, SG2, and SG3.

genes, whereas SG2 and SG3 showed more down-regulated genes of the oxidative phosphorylation pathway (Fig 3E and 3F).

## Gene copy number alteration analysis of T-PLL subgroups

Gene copy number data were available for a subset of 53 T-PLL patients (SG1: 11, SG2: 8, SG3: 34) to analyze whether T-PLL subgroup-specific alterations exist. Therefore, we visualized the gene copy number profiles in a heatmap according to the chromosomal order of genes with respect to their specific T-PLL subgroup assignments (Fig 5A). The different T-PLL subgroups showed similar copy number alterations. The majority of T-PLL patients had a characteristic deletion in combination with a duplication affecting chromosome 8. Deletions on chromosome 11 and duplications on chromosome 14 were also frequently observed. The copy number status of chromosome X was in accordance with the sex of patients.

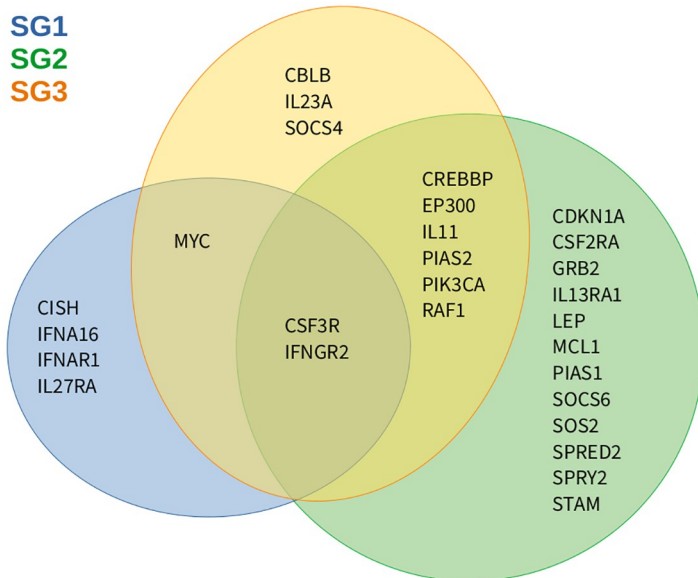

**Fig 4. Similarities and differences of up-regulated Jak-Stat signaling genes.** Venn diagram representing the overlap of up-regulated Jak-Stat signaling genes identified for the three T-PLL subgroups in comparison to normal controls. Colored circles represent the T-PLL subgroups (SG1: blue, SG2: green, SG3: orange).

A more focused search for subgroup-specific DNA copy number alteration patterns was done by a systematic comparison of the median gene copy number profiles of the three T-PLL subgroups (Fig 5B). These profiles highlighted again the previously observed frequent alterations, but also showed that the specific duplication affecting chromosome 8 was mainly present in SG1 and SG3, but only rarely observed in SG2. This duplication also recurrently affected known cancer census genes such as *CSMD3*, *COX6C*, *EXT1*, *FAM135B*, *MYC*, and *NDRG1* in SG1 and SG3 in more than 75% of patients. Duplications affecting chromosome 14 were more frequent in SG1 than in SG2 or SG3. In addition, duplications affecting chromosome 7 and 22 were more frequently observed in SG1 than in the other two subgroups. Chromosome 7 also showed a focal deletion in SG1 that affected *EZH2* in more than 50% of patients. Thus, despite strong similarities of genome-wide gene copy number alterations, several subgroup-specific differences were observed.

## Network-based analysis reveals potential major regulators that differ in their expression between the T-PLL subgroups

To identify potential major regulators that distinguish the three T-PLL subgroups, we integrated paired gene copy number and expression data available for 53 T-PLL patients to create putative gene regulatory networks associated with the 5,858 differentially expressed genes that differed between the subgroups. This was done using the R package regNet [30] (see Materials and methods section for details). The general idea of this approach is to predict the expression behavior of a gene based on its own copy number and the expression levels of other genes that best explain the observed expression behavior of the specific gene across the three T-PLL subgroups. This network inference was repeated 100 times based on randomly chosen subsets of the 53 T-PLL patients to determine links between genes that were found in the majority of the different network inference runs. Overall, the 100 learned networks contained relevant information to predict the expression behavior of genes in T-PLL significantly better than corresponding random networks of same complexity (S9 Fig, increase of median correlation:

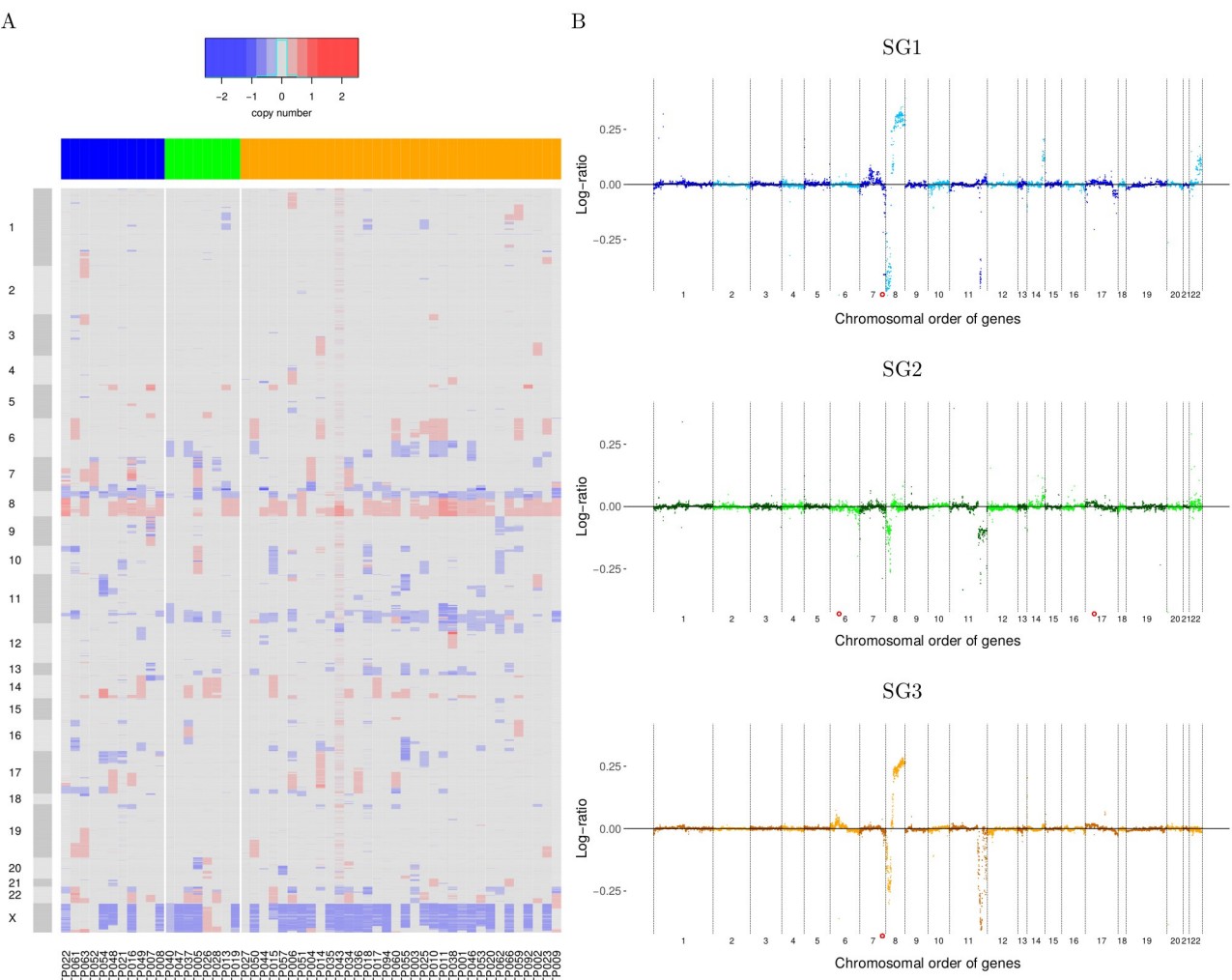

**Fig 5. Genome-wide gene copy number profiles of T-PLL samples.** A: Heatmap of gene copy number profiles of individual T-PLL samples highlighting deletions (blue), duplications (red), and genes with unchanged copy number (grey) in T-PLL in comparison to normal DNA. The columns represent the individual T-PLL samples of the three different T-PLL subgroups (SG1: blue, SG2: green, SG3: orange). Corresponding sample names are shown below the heatmap. The rows represent the genes in chromosomal order along the chromosomes. Chromosomes are highlighted by alternating grey shades left to the heatmap. B: Median gene copy number profiles of the three T-PLL subgroups. Chromosomes are separated by dotted vertical lines and the median copy number values of genes are shown by small colored dots. The color of these dots is altered with the chromosome according to the subgroup-specific base color to enable a better visual separation between chromosomes. Strong deviations of median values from zero indicate frequently observed deletions (negative log-ratios) and duplications (positive log-ratios) of genes in the corresponding subgroup. Red circles depict strongly negative median copy number log-ratios that were outside of the plotted range (*PRSS1*: SG1 and SG3, *HLA-DRB5* and *LGALS9C*: SG2).

0.4176, paired Wilcoxon signed rank test: $p < 2.2 \cdot 10^{-16}$). The learned networks were further used to create a consensus network by focusing on robust links between genes that were present in at least 75 of 100 networks (link cutoff: $q \leq 0.01$). To focus on genes with increased connectivity, consensus network modules consisting of genes that had at least two links to other genes in a module of at least three genes were visualized in Fig 6. This module representation includes 16 gene modules that consist of one up to five highly connected genes. Since these modules were derived from networks that were learned based on data of all three T-PLL subgroups, the module representation in Fig 6 also enables to see how the 41 included genes behave in their expression across the three subgroups in relation to the normal controls.

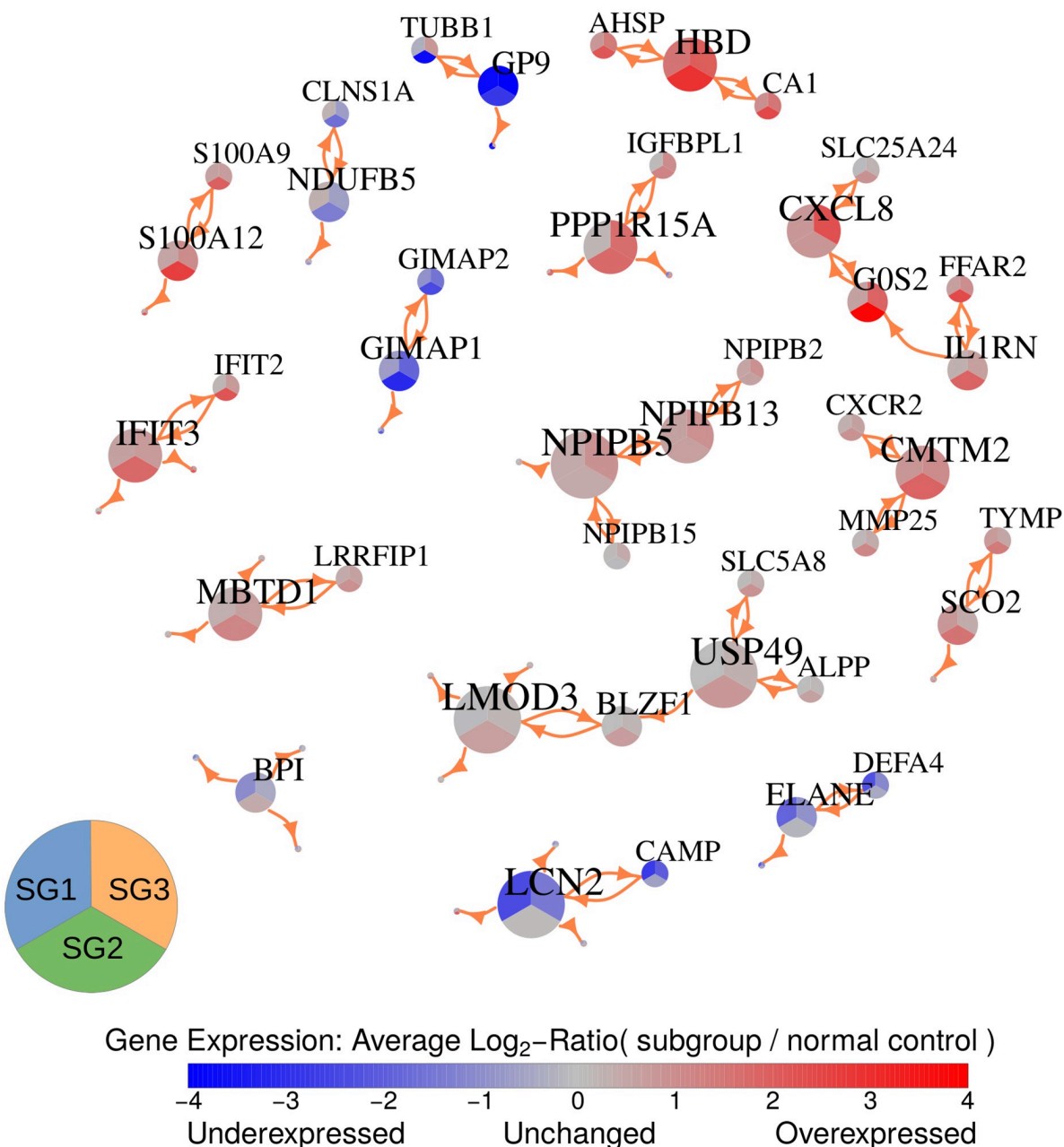

**Fig 6. Network-based visualization of the expression behavior of potential major regulators for each of the three T-PLL subgroups in comparison to normal control references.** The network represents gene modules highlighting genes that had at least two links to other genes in a module of at least three genes. Only links between genes that were present in at least 75 of the 100 learned networks for a link q-value cutoff of 0.01 were considered for the module selection. The labels of the nodes represent the corresponding gene names. The node size is proportional to the node-degree and each node is separated into three areas to represent the average expression difference of the underlying gene between the patients of the area-specific T-PLL subgroup (SG1, SG2, or SG3) and the control group (blue: down-regulated, grey: unchanged, red: up-regulated). All shown links between genes are potential activator links. These links can represent direct or indirect regulatory dependencies or may only show a correlation between expression levels of genes.

To better understand the cellular functions of these genes and to further analyze their associations with leukemia or other types of cancer, a functional gene annotation analysis based on [31] in combination with an in-depth literature search was done. Table 1 briefly summarizes the results of this analysis. More details to cancer-associated functions of individual genes

**Table 1. Summary of annotation and literature analysis of predicted potential major regulators.**

| Mod | Gene | Chr | SG1 | SG2 | SG3 | Cellular function | Ref |
|---|---|---|---|---|---|---|---|
| 1 | LMOD3 | 3 | = | + | = | organization of actin filaments | – |
| | USP49 | 6 | = | + | = | ubiquitin specific peptidase | [32] |
| | BLZF1 | 1 | = | + | = | protein transport to cell surface | [33] |
| | ALPP | 2 | = | + | = | alkaline phosphatase, stem cell diff. | [34]* |
| | SLC5A8 | 12 | = | + | = | sodium-coupled solute transporter | [35] |
| 2 | NPIPB5 | 16 | = | = | + | nuclear pore interacting protein | – |
| | NPIPB13 | 16 | + | + | + | nuclear pore interacting protein | – |
| | NPIPB2 | 16 | = | = | + | nuclear pore interacting protein | – |
| | NPIPB15 | 16 | = | = | + | nuclear pore interacting protein | – |
| 3 | LCN2 | 9 | – | = | = | transporter, apoptosis, immunity | [36]* |
| | CAMP | 17 | – | = | – | chemotaxis, immunity, inflammation | – |
| 4 | CXCL8 | 4 | = | + | + | chemokine, inflammation | [37]* |
| | G0S2 | 1 | = | + | + | promotes apoptosis | [38]* |
| | IL1RN | 2 | = | + | + | immunity, inflammation | [39]* |
| | FFAR2 | 19 | + | + | + | G protein receptor, immunity | [40]* |
| | SLC25A24 | 1 | = | + | = | solute carrier, protect. oxidative stress | [41] |
| 5 | IFIT3 | 10 | = | + | = | inhibition, migration and proliferation | [42] |
| | IFIT2 | 10 | = | + | = | interferon induced, promotes apoptosis | [43] |
| 6 | HBD | 11 | + | + | + | hemoglobin subunit delta | – |
| | AHSP | 16 | = | + | + | chaperone, erythroid cell development | [44]* |
| | CA1 | 8 | = | + | + | carbonic anhydrase, hydration of $CO_2$ | [45] |
| 7 | CMTM2 | 16 | + | + | + | chemokine-like factor | [46] |
| | CXCR2 | 2 | = | + | = | G-protein receptor, neutrophil migration | [47]* |
| | MMP25 | 16 | = | + | = | matrix proteinase, invasion, metastasis | [48] |
| 8 | PPP1R15A | 19 | = | + | + | growth arrest, apoptosis | [49]* |
| | IGFBPL1 | 9 | = | + | + | insulin like growth factor binding protein | [50] |
| 9 | MBTD1 | 17 | = | + | = | polycomb group protein, epigenetic reg. | [51]* |
| | LRRFIP1 | 2 | = | + | = | transcriptional repressor | [52] |
| 10 | ELANE | 19 | – | = | = | elastase, serine protease, immunity | [53]* |
| | DEFA4 | 8 | – | = | = | defensin, immunity | – |
| 11 | GIMAP1 | 7 | = | – | – | lymphocyte survival, diff. T helper cells | [54] |
| | GIMAP2 | 7 | = | – | – | immuno-associated GTP-binding protein | [55]* |
| 12 | NDUFB5 | 3 | = | – | – | subunit mitoch. NADH dehydrogenase | – |
| | CLNS1A | 11 | = | – | – | involved in splicing | – |
| 13 | BPI | 20 | – | = | = | protection, detoxification | – |
| 14 | S100A12 | 1 | = | + | = | calcium binding protein, immunity, infla. | [56] |
| | S100A9 | 1 | = | + | = | immunity, inflammation, apoptosis | [57] |
| 15 | GP9 | 3 | – | – | – | membr. glycoprotein platelets, adhesion | [58] |
| | TUBB1 | 20 | – | – | – | beta tubulin, platelets, megakaryocytes | [59] |
| 16 | SCO2 | 22 | + | + | = | cytochrome c oxidase | [60] |
| | TYMP | 22 | + | + | = | angiogenic factor, blood vessel integrity | [61] |

Genes are listed according to the network module (column: Mod) to which they were grouped. The chromosome and the expression behavior ('–': down-regulated, '=' unchanged, '+': up-regulated) in each of the three T-PLL subgroups (columns: SG1, SG2, SG3) in comparison to normal controls are provided for each gene (S4 Table, $q \leq 0.05$). Cellular functions were obtained from [31]. Cancer-relevant publications of individual genes are listed and those in the context of leukemias are marked by '*' (column: Ref). See S1 Appendix for details to the literature analysis of individual genes.

from the literature search are provided in S1 Appendix. Overall, the functional gene annotation analysis allowed to group the majority of genes into five more general categories (Table 1): (i) genes involved in immune responses (*CAMP*, *CXCL8*, *CXCR2*, *DEFA4*, *ELANE*, *FFAR2*, *IFIT2*, *IFIT3*, *IL1RN*, *LCN2*, *S100A9*, *S100A12*), (ii) genes involved in angiogenesis or encoding of blood cell-specific components (*AHSP*, *CA1*, *ELANE*, *GIMAP1*, *GIMAP2*, *GP9*, *HBD*, *TUBB1*, *TYMP*), (iii) genes involved in cellular respiration or oxidative stress (*CA1*, *NDUFB5*, *SCO2*, *SLC25A24*), (iv) genes involved in cell proliferation, apoptosis, migration or invasion (*G0S2*, *IFIT2*, *IFIT3*, *LCN2*, *MMP25*, *PPP1R15A*, *S100A9*), and (v) genes interacting with nuclear pore complexes (*NPIPB2*, *NPIPB5*, *NPIPB13*, *NPIPB15*). Moreover, 30 of 41 genes in Table 1 have already been reported to play important roles in different types of cancer (S1 Appendix). Several of them have also been reported in context of different leukemias (Table 1: *AHSP*, *ALPP*, *CXCL8*, *G0S2*, *LCN2*, *IL1RN*, *FFAR2*, *CXCR2*, *PPP1R15A*, *MBTD1*, *ELANE*, *GIMAP2*).

Considering the expression behavior of the genes in the modules, most gene modules in Table 1 consisted of genes that mainly showed the same expression behavior in a specific T-PLL subgroup. Only two genes were down-regulated (*GP9*, *TUBB1*) and only four genes were up-regulated (*NPIPB13*, *FFAR2*, *HBD*, *CMTM2*) across all three T-PLL subgroups, whereas the other genes showed expression patterns that differed at least in one of the three subgroups. Several genes were unchanged in SG1 but up-regulated in SG2 and SG3 in comparison to normal controls (Table 1 module 4: *CXCL8*, *G0S2*, *IL1RN*; module 6: *AHSP*, *CA1*; module 8: *PPP1R15A*, *IGFBPL1*). Similarly, some genes were unchanged in SG1 but down-regulated in SG2 and SG3 (Table 1 module 11: *GIMAP1*, *GIMAP2*; module 12: *NDUFB5*, *CLNS1A*). Four genes were exclusively down-regulated in SG1 but unchanged in SG2 and SG3 in comparison to normal controls (Table 1 module 3: *LCN2*; module 10: *DEFA4*, *ELANE*; module 13: *BPI*). Further, several genes were exclusively up-regulated in SG2 but unchanged in SG1 and SG3 in comparison to normal controls (Table 1 module 1: *ALPP*, *BLZF1*, *LMOD3*, *SLC5A8*, *USP49*; module 5: *IFIT2*, *IFIT3*; module 7: *CXCR2*, *MMP25*; module 14: *S100A9*, *S100A12*). Three genes of the family of nuclear pore complex interacting proteins were exclusively up-regulated in SG3 but unchanged in SG1 and SG2 (Table 1 module 2: *NPIPB5*, *NPIPB2*, *NPIPB15*).

Thus, in relation to the reported functions, several of the predicted genes with increased network connectivity could potentially contribute to the manifestation of the observed differences between the three T-PLL subgroups at the level of individual genes, cellular pathways and regulatory networks to alter cell proliferation and immune responses in a subgroup-specific manner.

## Occurrence of the revealed T-PLL gene expression subgroups in other smaller patient cohorts

To further validated the existence of the three revealed T-PLL subgroups, RNA-seq data of 41 T-PLL patients from the Herling laboratory were considered. This cohort contained 13 patients that were also included in our microarray data set. Each of these 13 patient-specific RNA-seq samples always had the strongest correlation with the gene expression profile of its corresponding patient-specific microarray sample (). Thus, the sample-specific gene expression patterns were conserved across the different experimental platforms and also the subgroup assignments of these 13 patients were stable. In addition, the T-PLL subgroup label of each of the other 28 patients was determined by transferring the subgroup label of the microarray sample that showed the strongest correlation to the corresponding RNA-seq sample. Overall, the resulting distribution of the subgroups across the 41 patients was very similar to those

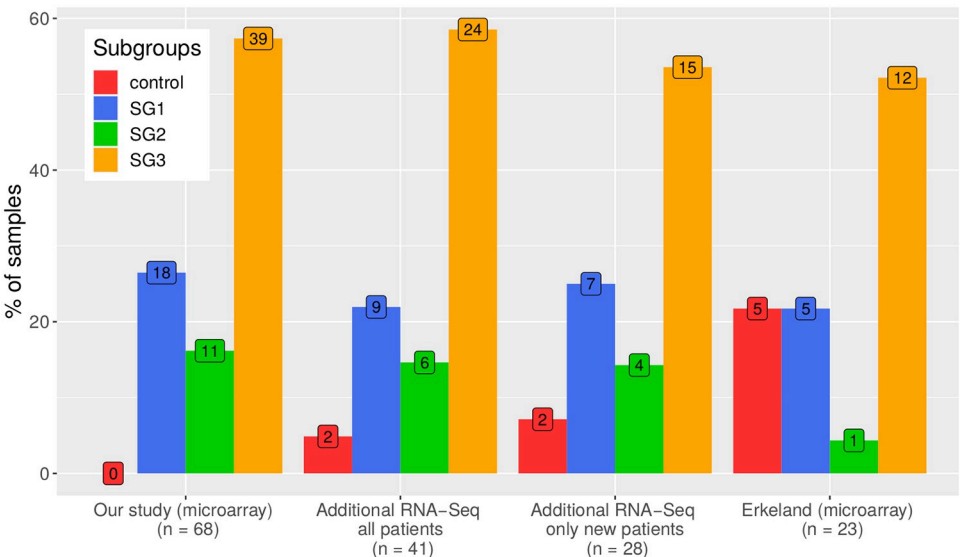

**Fig 7. Occurrences of the three revealed T-PLL subgroups in two other T-PLL cohorts.** Barplots represent the percentages of T-PLL patient samples that were assigned to the three revealed T-PLL gene expression subgroups (SG1, SG2, SG3) or the normal control group (control). The first group of colored barplots on the left side represents the subgroup distribution of the T-PLL samples in our microarray-based study, the two groups of colored barplots in the middle represent the subgroup distribution of T-PLL samples in the RNA-seq data from the Herling lab including overlapping patients (all patients) or excluding patients that overlap with our microarray-based study (only new patients), and the fourth group of colored barplots on the right side represents the subgroup distribution for the T-PLL microarray samples from [25]. The numbers in the colored boxes additionally provide the exact number of samples in each subgroup.

of our microarray data set (Fig 7) and did not differ significantly (Fisher's exact test: $p = 0.39$). The subgroup distribution was also well preserved when the 13 overlapping patients were removed (Fig 7, Fisher's exact test: $p = 0.23$). This emphasizes the robustness of the finding and suggests that independent T-PLL patients also reflect the three identified T-PLL subgroups.

Moreover, also recently published microarray gene expression profiles of 23 T-PLL patients by Erkeland *et al.* [25] were considered to analyze how these patients behave in relation to the three T-PLL subgroups. Most patients were assigned to subgroup SG3 followed by assignments to SG1, whereas only one patient was assigned to SG2 (Fig 7). Samples of five patients were assigned to the normal control group, which could potentially be related to a reduced content of leukemic cells in these samples. This is supported by significantly lower white blood cell counts for these five samples compared to samples of patients that were assigned to SG1 or SG3 (S11 Fig). Excluding these potentially spurious samples, the distribution of the subgroups among the remaining 18 patients did again not significantly differ from those of our initial microarray data set (Fisher's exact test: $p = 0.65$). This similarity of the subgroup distributions is further supported by the additional observation that patients of SG3 tend to survive longer than patients of SG1 (S12 Fig), which was also found for SG1 and SG3 patients of our cohort (S4A Fig).

All these findings for these two T-PLL validation cohorts suggest that the T-PLL gene expression subgroups could potentially be relevant to better stratify T-PLL patients in general.

## Discussion

We performed an in-depth analysis of one of the largest publicly available gene expression data sets of T-PLL patients from [17] with the goal to identify potentially existing T-PLL

subgroups for this rare leukemia. Hierarchical clustering of the gene expression profiles enabled us to predict three robust T-PLL gene expression subgroups that have not been reported so far. Interestingly, the predicted T-PLL subgroup SG1 was co-clustered together with the normal references in a joint subtree, whereas the other two subgroups SG2 and SG3 were part of a separate subtree (Fig 1). The stability of these subgroups was confirmed by bootstrapping approaches and a transfer to independent T-PLL patients from two other validation cohorts also supported the existence of these subgroups.

To systematically characterize similarities and differences between the three revealed T-PLL subgroups, we performed a comparative analysis of the predicted gene expression alterations. We generally observed more down- than up-regulated genes in each subgroup in comparison to the normal controls (Fig 2). Expression alterations of gene sets of essential genes and cancer census genes were significantly enriched in each subgroup, but many of these and also other genes were only exclusively altered in a specific subgroup. Thus, subgroup-specific expression alterations of individual genes were more prevalent than common expression alterations of genes in all three or between two of three subgroups. Nevertheless, this indicates that a core set of altered genes, which distinguishes all three T-PLL subgroups from normal controls, exists. This includes for example the commonly up-regulated potential major regulators *CMTM2* [46], *FFAR2* [40], and *NPIPB13* and the commonly down-regulated potential major regulators *GP9* [58] and *TUBB1* [59]. Such commonly altered genes could provide a basis for the development of targeted treatment strategies from which all subgroups may profit. However, it is also important to better understand how additional subgroup-specific gene expression alterations influence T-PLL development and progression. These alterations may for example contribute to the observed subgroup-specific expression behavior of the T-PLL drivers *TCL1A* and *ATM* (S7 Fig) and may thereby influence the development of specific subgroups.

Differences between the three T-PLL subgroups were also observed at the level of signaling and metabolic pathways (Fig 3). Again more down- than up-regulated pathway genes were observed for each subgroup. In accordance with the initial hierarchical clustering, the pathway alteration profiles of SG2 and SG3 were more similar to each other than those of SG1. Overall, each subgroup had its characteristic pathway expression profile. Specific enrichments of altered pathway genes were exclusively found for each subgroup comprising down-regulated cytokine receptor interaction, apoptosis, and focal adhesion genes in SG1, up-regulated MAPK signaling and apoptosis genes and down-regulated nucleotide excision repair genes in SG2, and up-regulated nucleotide excision repair genes in SG3. The only shared enrichments were down-regulated pyruvate and the purine metabolism pathway genes in SG2 and SG3. Such observations can be of importance for the development of targeted treatment strategies. Therefore, we also performed a detailed analysis of the Jak-Stat signaling pathway (Fig 4), which is important in T-cell-related cancers and one of the potential promising targets for the development of novel T-PLL treatment strategies [9, 22]. Several commonalities but also important differences in the expression of Jak-Stat signaling pathway genes between the three T-PLL subgroups were revealed. This may influence the efficiency of targeted Jak-Stat treatments in a subgroup-specific manner.

In addition to the gene expression profiles, we also analyzed gene copy number profiles that were available for 53 of the 68 considered T-PLL patients. Strong similarities of gene copy number profiles of the three T-PLL subgroups were globally observed (Fig 5). The majority of T-PLL patients showed recurrent deletions affecting the chromosomal arms 8p and 11q. Also recurrent duplications affecting the chromosomal arm 8q were frequently observed. These observations are in good accordance with prior studies [17, 23, 62, 63]. Generally, deletions affecting the chromosomal arm 11q are usually linked to a monoallelic loss of the tumor suppressor *ATM* (11q22.3), which contributes to T-PLL development by a deregulation of DNA

repair [62–64]. The observed subgroup-specific expression of ATM, which was significantly down-regulated in SG2 and SG3 but not in SG1 in comparison to normal controls (S11 Fig), could contribute to the development of individual T-PLL subgroups. Further, duplications of the chromosomal arm 8q can contribute to increased expression levels of the proto-oncogene *MYC* (8q24.21), which has been identified as a target of Jak-Stat signaling [65, 66], but also other genes on chromosome 8q like *AGO2* (8q24.3) [17], which is involved in RNA interference, could contribute to T-PLL development. Overall, duplications affecting the chromosomal arm 8q were mainly observed for T-PLL patients from SG1 and SG3 but only rarely found in SG2. This could for example influence the expression behavior of the Jak-Stat signaling pathway via the *MYC*-axis and thereby contribute to the development of T-PLL subgroups. Further, duplications affecting chromosomes 7 and 22 were more frequently observed in SG1 than in SG2 or SG3. Globally, the observed recurrent DNA copy number alterations affect hundreds of genes. It is likely that at least several of them may contribute to T-PLL development and treatment response. Additional studies are required to predict such genes. A promising strategy would be the analysis of genome-wide T-PLL-specific gene regulatory networks with the help of network flow algorithms to identify subgroup-specific driver candidates [5]. The value of such approaches has already been demonstrated for other types of cancer [67–69]. Additional studies are required to transfer such an approach to T-PLL.

To identify potential major regulators along with their subgroup-specific expression behavior, we learned gene regulatory networks associated with the observed expression changes between the three T-PLL subgroups (Fig 6). Most of the 41 revealed genes with increased network connectivity have been reported to be involved in different types of cancer and several of them have been shown to be important in different types of leukemia (Table 1, *ALPP* [34], *AHSP* [44], *CXCL8* [37], *CXCR2* [47], *ELANE* [53], *FFAR2* [40], *G0S2* [38], *GIMAP2* [55], *IL1RN* [39], *LCN2* [36], *MBTD1* [51], *PPP1R15A* [49]). Thus, it is likely that at least some of these genes may also contribute to T-PLL development and that their subgroup-specific expression behavior may contribute to the globally observed gene expression differences of the T-PLL subgroups. Further, considering the functional annotations of these 41 genes (Table 1), altered immune responses, differences in angiogenesis or the expression of blood cell-specific components, altered levels of cellular respiration and oxidative stress, differences in cell proliferation, apoptosis, migration and invasion, and alterations of the cellular transport system between nucleus and cytoplasm may at least in part distinguish the three revealed T-PLL subgroups.

Interestingly, several of the network gene modules contained putative oncogenes and tumor suppressors whose expression levels were commonly altered in the same direction. For example, the putative oncogene *IFIT3* [42] and the putative tumor suppressor *IFIT2* [43] were both exclusively up-regulated in SG2, the putative oncogene *TYMP* [61] and the putative tumor suppressor *SCO2* [60] were both up-regulated in SG1 and SG2, or the putative oncogene *PPP1R15A* [49] and the putative tumor suppressor *IGFBPL1* [50] were both up-regulated in SG2 and SG3. Such an expression behavior of actually counteracting genes has already been reported for other types of cancer (e.g. [43, 68]). A balance shift of the expression levels of such counteracting genes could potentially influence the proliferation of T-PLL cells within specific subgroups. Therefore, especially up-regulated oncogenes could potentially represent promising targets for drug-based interventions.

Overall, our study demonstrated that T-PLL patients can be stratified into three distinct gene expression subgroups. These three T-PLL subgroups have the potential to contribute to an improved molecular stratification of T-PLL patients. Although, differences in survival were not observed between the subgroups, their specific molecular characteristics that we identified could potentially be of great relevance for the design and analysis of drug screens and future

developments of urgently needed targeted treatment strategies and corresponding clinical studies. The revealed T-PLL subgroups might be associated with different responses to treatments. Such knowledge could provide an important basis for patient-specific treatment decisions. Additional experimental studies are required to analyze the response behavior of the revealed subgroups in drug screens. Subgroup-specific pathomechanisms could potentially be revealed by gene perturbation experiments that target some of the predicted potential major regulators.

## Materials and methods

### Gene expression data of T-PLL and normal control samples

Normalized microarray gene expression data of tumor T-cells of 70 T-PLL patients and normal control CD3$^+$ pan T-cells of 10 healthy donors were downloaded from the Gene Expression Omnibus (GEO) data base (GSE107513) [17]. Purified T-cells obtained from peripheral blood of T-PLL patients and of healthy donors formed the basis of the gene expression profiling. These T-cells were derived by gradient centrifugation followed by magnetic-bead based cell enrichment reaching a purity of more than 95% [17]. Technical replicates measured for four T-PLL patients (TP002, TP003, TP059, TP063) were averaged. For genes with more than one probe, corresponding average gene-specific probe-based gene expression levels were computed for each sample. The resulting gene expression data set comprised expression levels of 17,970 genes. Recently updated clinical data of the T-PLL patients are available in S1 Table. Most samples were from untreated T-PLL patients at diagnosis (58 of 70) and 12 samples were from patients in relapse after a previous treatment (S1 Table). The gene expression data set is available in S2 Table.

### Identification of T-PLL subgroups

Hierarchical clustering of gene expression profiles was performed using R, applying $1-r$ with $r$ denoting the Pearson correlation as distance measure in combination with Ward's linkage criterion (ward.D2) [70]. Hierarchical clustering initially considers each expression profile as a separate cluster and then iteratively repeats the following two steps until all clusters are merged together: (i) identification of the two clusters with the smallest distance followed by (ii) merging of these two clusters into a joint cluster. These iterative merging steps allow to identify hierarchical relationships between the clusters that are stored in a tree-structure referred to as dendrogram. The resulting dendrogram was automatically cut into four subtrees after visual inspection. One of these subtrees consisted of the normal control samples, whereas the other three subtrees represented the newly revealed T-PLL subgroups (SG1, SG2, SG3). Note that we excluded the T-PLL samples TP032 and TP033 from the analyses, because they had expression profiles very similar to those of the normal control cells and a re-evaluation of the immunophenotypic and clinical representation of both samples did not confidently verify them as T-PLL. Assignments of individual T-PLL patients to the three T-PLL subgroups are provided in S3 Table. The R package pvclust was used with standard bootstrap settings to assess the robustness of the obtained hierarchical clustering by bootstrapping of genes [27]. The robustness of individual clusters was quantified by approximately unbiased (AU) p-values averaged over 10,000 individual runs (S1 Fig). The stability of the subgroup assignments was further validated by repeating the hierarchical clustering for subsets of the initial data set by randomly removing one up to ten T-PLL patient samples. The four resulting major clusters of each repeated hierarchical clustering were paired to the four initially obtained clusters (control, SG1, SG2, SG3) based on the majority of overlapping samples. This allowed to determine the number of correctly clustered samples for each run (S2 Fig).

## Survival analysis of T-PLL subgroups

Information about time to death for patients with reported status 'died of disease' (DOD) or time to last follow-up for patients with reported status 'alive with disease' (AWD) were updated since the publication of the initial study by [17] and summarized together with other clinical information for the considered T-PLL patients (S1 Table). The AWD status was considered as non-informative censoring event. Kaplan-Meier curves were created and log-rank tests were performed using the R package survival [71] to compare the survival of patients in the three different T-PLL subgroups and to compare the survival of patients based on sex independent of their subgroup assignments.

## Identification of differentially expressed genes in T-PLL subgroups

Differential gene expression analysis was done for each of the three revealed T-PLL subgroups in comparison to the normal control cells following limma's standard workflow [72]. Differentially expressed genes were selected using an FDR-adjusted p-value (q-value) [73] cutoff of 0.05. The results of the differential gene expression analysis are provided in S4 Table. Volcano plots and Venn diagrams were used to visualize the results.

## Pathway and gene enrichment analysis of T-PLL subgroups

Basic cancer-relevant gene annotation categories were taken from [67]. The underlying signaling and metabolic pathways from KEGG/Reactome were updated by more recent information from ConsensusPathDB [74]. The set of cancer census genes was updated to a more recent version (release March 2020). All utilized pathway and gene annotations are provided in S5 Table. Pathway and gene annotations were considered to test for an enrichment of differentially expressed genes in each specific annotation category. For each comparison of one of the three T-PLL subgroups to the normal control cells, the number of differentially expressed genes in each annotation category was counted separately for down- and up-regulated genes (S4 Table, $q \leq 0.05$). Significance of enrichment per annotation category was determined using Fisher's exact test. Correction for multiple testing was done by computing FDR-adjusted p-values (q-value) [73] using the R function p.adjust. Bar plots were used to represent the results and to label significantly enriched categories.

## Gene copy number data analysis of T-PLL subgroups

DNA copy number profiles of 53 T-PLL patients that were included in our gene expression data set had been measured by [17]. We downloaded these data from GEO (GSE107513). The obtained copy number $\log_2$-ratios quantify for each genomic region the copy number in T-PLL in relation to normal control DNA. We sorted the $\log_2$-ratios of each patient by their chromosomal probe locations and segmented the resulting chromosome-specific copy number profiles into regions of constant copy number using DNACopy [75]. Based on that, copy number values of 17,671 genes were determined for each T-PLL sample by mapping the chromosomal locations of genes to the obtained chromosomal segments as described in [67]. The gene copy number data of all T-PLL samples are available in S6 Table. We utilized a heatmap to visualize the individual gene copy number profiles of samples of the three T-PLL subgroups. This enabled to identify specific chromosomal regions affected by gene deletions and duplications in individual samples. Frequent gene deletions and duplications were further determined for each T-PLL subgroup based on the corresponding median gene copy number alteration profile.

## Network-based prediction of potential major regulators distinguishing T-PLL subgroups

All 5,858 differentially expressed genes predicted in the comparison of the three T-PLL subgroups to the normal control samples at the q-value cutoff of 0.05 (S4 Table) were used to learn a gene regulatory network to identify potential major regulators that differ in their expression behavior between the T-PLL subgroups. The expression of each gene was modeled as a linear combination of its own gene copy number and the expression values of all other genes. The parameters of the underlying linear models were computed using the R package regNet [30], which uses lasso regression [76] in combination with a significance test for lasso [77], to determine the most relevant predictors for each gene-specific linear model. Depending on the sign of the learned parameter, a selected predictor can either represent a potential activator (positive sign) or a potential inhibitor (negative sign) of the considered gene. Since each gene can be selected as a potential regulator of other genes, a global network is fully determined by the gene-specific linear models. Additional details to the underlying concept are provided in [30, 67]. Closely related approaches have successfully been applied to gene expression signatures of other cancers [78–81]. Following [80], this network inference was repeated 100 times based on randomly created training sets that comprised 35 samples representing two thirds of the 53 T-PLL samples for which gene expression and copy number data were available. The remaining one third of T-PLL samples (18 of 53), which were not in the training set, were considered as independent network-specific test set. The prediction quality of each learned network was determined based on its corresponding test set by computing the correlation between predicted and originally measured gene expression levels. These computations were done for each originally learned network including network links at a q-value cutoff of 0.01 and for its corresponding 10 random network instances of same complexity created by degree-preserving network permutations. The prediction quality of each gene was further averaged across the networks to compare the prediction quality of the originally learned networks to those of the corresponding random networks. The prediction quality of the individual genes was generally high and significantly shifted into the positive range (S9 Fig, median correlation of red distribution: 0.426, one-sample Wilcoxon signed rank test: $p < 2.2 \cdot 10^{-16}$) and also significantly better than those of corresponding random networks of same complexity (S9 Fig, median difference of red vs. grey distribution: 0.4176, paired Wilcoxon signed rank test: $p < 2.2 \cdot 10^{-16}$). Thus, the learned networks contained relevant information to predict T-PLL gene expression behavior. Further, to obtain an integrative view of the gene expression differences between the three T-PLL subgroups, the Fruchterman and Reingold layout algorithm of the R package igraph was used to visualize the links between genes that were present in at least 75 of 100 networks at a link q-value cutoff of 0.01 (S7 Table) displaying groups of highly connected genes (gene modules) with at least two links in a module of at least three genes.

## Validation of revealed T-PLL gene expression subgroups in other patient cohorts

To validate the existence of the three revealed T-PLL subgroups, gene expression data of two other patient cohorts were considered. The first validation cohort consisted of 41 T-PLL patients for which transcriptome sequencing (RNA-seq) data were generated in the Herling lab. The RNA-seq data was processed as described in [82] and the resulting normalized gene expression levels were made comparable to our initially considered microarray data set from [17] by a batch correction using the ComBat function of the R package sva with standard settings [83]. The obtained RNA-seq validation data set and the corresponding adjusted

microarray data set of our main study are provided in S8 Table. Each of the 41 patient-specific T-PLL gene expression profiles was used to determine its similarity to each sample in our microarray data set by computing corresponding pairwise Pearson correlation coefficients. Each new sample was then assigned to the known subgroup (control, SG1, SG2, SG3) of the microarray sample with the strongest positive correlation. This allowed to analyze the distribution of the three revealed T-PLL subgroups among the new patients. Since 13 of the 41 T-PLL patients were also part of the microarray data set, this classification also allowed to analyze if each of this 13 RNA-seq samples was assigned to its corresponding microarray sample. Similarly, a second validation cohort of 23 T-PLL patients from Erkeland *et al.* [25] was prepared to analyze the distribution of the subgroups among the patients. Processed microarray data was downloaded from GEO (GSE147930). The probe-based expression levels were mapped to the genes followed by a batch correction with ComBat [83] to ensure the comparability to our microarray data set. The microarray validation data set and the corresponding adjusted microarray data set of our main study are provided in S9 Table. Subgroup assignments of the 23 new T-PLL patients were made as described for the RNA-seq validation cohort.

## Supporting information

**S1 Fig. Stability analysis of hierarchical clustering based on bootstrapping of genes.**
(PDF)

**S2 Fig. Stability of T-PLL subgroup assignments based on bootstrapping of patients.**
(PDF)

**S3 Fig. Associations between TCL1A protein expression and *TCL1A* gene expression of T-PLL patients.**
(PDF)

**S4 Fig. Kaplan-Meier analyses of T-PLL subgroups.**
(PDF)

**S5 Fig. Gene annotation analysis of differentially expressed genes of T-PLL subgroups.**
(PDF)

**S6 Fig. Heatmap-based clustering considering differentially expressed genes.**
(PDF)

**S7 Fig. Subgroup-specific *TCL1A* and *ATM* expression.**
(PDF)

**S8 Fig. Venn diagrams of subgroup-specific pathway overlaps.**
(PDF)

**S9 Fig. Network prediction quality.**
(PDF)

**S10 Fig. Pairwise correlations between microarray and RNA-seq gene expression profiles.**
(PDF)

**S11 Fig. White blood cell counts for T-PLL patients from Erkeland assigned to the three T-PLL subgroups or the normal control group.**
(PDF)

**S12 Fig. Kaplan-Meier analysis of T-PLL patients from Erkeland assigned to subgroup SG1 and SG3.**
(PDF)

**S1 Table. Clinical data of T-PLL patients.**
(XLSX)

**S2 Table. Gene expression data considered for T-PLL subgroup discovery.**
(TSV)

**S3 Table. Subgroup assignments of T-PLL patients.**
(TSV)

**S4 Table. Subgroup-specific differential gene expression analysis.**
(XLSX)

**S5 Table. Utilized pathway and gene annotations.**
(XLS)

**S6 Table. Gene copy number data of T-PLL patients.**
(TXT)

**S7 Table. Network connectivity table.**
(XLSX)

**S8 Table. RNA-seq validation data from Herling laboratory.**
(TSV)

**S9 Table. Microarray validation data from Erkeland.**
(TSV)

**S1 Appendix. Summary of cancer-associated literature analysis of predicted potential major regulators.**
(PDF)

## Author Contributions

**Conceptualization:** Michael Seifert.

**Data curation:** Nathan Mikhaylenko, Linus Wahnschaffe, Michael Seifert.

**Formal analysis:** Nathan Mikhaylenko, Michael Seifert.

**Funding acquisition:** Marco Herling, Ingo Roeder, Michael Seifert.

**Methodology:** Nathan Mikhaylenko, Michael Seifert.

**Project administration:** Marco Herling, Ingo Roeder, Michael Seifert.

**Resources:** Marco Herling, Ingo Roeder, Michael Seifert.

**Supervision:** Marco Herling, Ingo Roeder, Michael Seifert.

**Writing – original draft:** Nathan Mikhaylenko, Michael Seifert.

**Writing – review & editing:** Linus Wahnschaffe, Marco Herling, Ingo Roeder.

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
