## [Decision Letter · Decision Letter 0]

22 Jul 2022

PONE-D-22-16001Computational gene expression analysis reveals distinct molecular subgroups of T-cell prolymphocytic leukemiaPLOS ONE

Dear Dr. Seifert,

Thank you for submitting your manuscript to PLOS ONE. After careful consideration, we feel that it has merit but does not fully meet PLOS ONE’s publication criteria as it currently stands. Therefore, we invite you to submit a revised version of the manuscript that addresses the points raised during the review process by Reviewer #1.

We look forward to receiving your revised manuscript.

Kind regards,

Francesco Bertolini, MD, PhD

Academic Editor

PLOS ONE

Journal Requirements:

Reviewers' comments:

Reviewer's Responses to Questions

**Comments to the Author**

1. Is the manuscript technically sound, and do the data support the conclusions?

Reviewer #1: Partly

Reviewer #2: Yes

2. Has the statistical analysis been performed appropriately and rigorously? 

Reviewer #1: Yes

Reviewer #2: Yes

3. Have the authors made all data underlying the findings in their manuscript fully available?

Reviewer #1: Yes

Reviewer #2: Yes

4. Is the manuscript presented in an intelligible fashion and written in standard English?

Reviewer #1: Yes

Reviewer #2: Yes

5. Review Comments to the Author

Reviewer #1: TPLL is a rare disease with a very poor prognosis. This article identifies 3 subgroups of TPLL according to their transcriptional profile from data base.

The manuscript is clearly written and structured

Major comments:

The manuscript lacks mechanistic data. Have there been verifications of the main candidate genes identified by another technique?

There are duplicate patients between the training and validation series. They should be removed from the analysis

Are the patients at diagnosis or in relapse? The table in the supplemental data seems to indicate more patients in follow-up. Specify the number and lines of treatment received.

Could this recruitment heterogeneity have an impact on the molecular classification identified?

Hyperleukocytosis is not necessarily high. Was there a sorting of tumor cells or a study of the cell composition to avoid contamination by normal cells?

Specify the exact nature of the normal T cell population of reference

What clinical interest could these identified signatures have?

How is the expression of TCL1a defined? Is there a correlation with GEP?

Reviewer #2: Dear Editor,

Thank you for the opportunity to review the manuscript entitled: “Computational gene expression analysis reveals distinct molecular subgroups of T-cell prolymphocytic leukemia” by Mikhaylenko et al.

The study described the computational analysis of 68 gene expression profiles of one of the largest currently existing T-PLL patient cohorts. They identified three T-PLL subgroups: SG1, SG2 and SG3 that differed for genes that were either up-regulated or down-regulated. They also compared these profiles with putative normal counterparts. Differences between the three T-PLL subgroups were also observed at the level of signaling and metabolic pathways.

Overall the study is well organized, clear in its structure and with some interesting conclusions. The analysis is rigorous and well-conducted. Figures are clear. The data were taken from Schrader A et al, Nat Commun. 2018 and thoroughly re-analyzed. While Schrader A et al also identified 4 groups (1 mostly normal samples, and 3 T-PLL groups), their analysis was not as deep as in the present manuscript. Thus, some novelty and new concepts are described in the present manuscript that could be useful for researchers working with T-PLL.

Technically, there are not major revision needed as the analysis are overall well-conducted.

6. PLOS authors have the option to publish the peer review history of their article (what does this mean?). If published, this will include your full peer review and any attached files.

Reviewer #1: No

Reviewer #2: No

---

## [Author Response · Author response to Decision Letter 0]

1 Aug 2022

Detailed responses to reviewer comments are provided in the attached PDF file: ResponseToReviewerComments.pdf

---

## [Decision Letter · Decision Letter 1]

30 Aug 2022

Computational gene expression analysis reveals distinct molecular subgroups of T-cell prolymphocytic leukemia

PONE-D-22-16001R1

Dear Dr. Seifert,

We’re pleased to inform you that your manuscript has been judged scientifically suitable for publication and will be formally accepted for publication once it meets all outstanding technical requirements.

Kind regards,

Francesco Bertolini, MD, PhD

Academic Editor

PLOS ONE

Additional Editor Comments (optional):

Reviewers' comments:

Reviewer's Responses to Questions

**Comments to the Author**

1. If the authors have adequately addressed your comments raised in a previous round of review and you feel that this manuscript is now acceptable for publication, you may indicate that here to bypass the “Comments to the Author” section, enter your conflict of interest statement in the “Confidential to Editor” section, and submit your "Accept" recommendation.

Reviewer #1: (No Response)

Reviewer #2: All comments have been addressed

2. Is the manuscript technically sound, and do the data support the conclusions?

Reviewer #1: Yes

Reviewer #2: Yes

3. Has the statistical analysis been performed appropriately and rigorously? 

Reviewer #1: Yes

Reviewer #2: Yes

4. Have the authors made all data underlying the findings in their manuscript fully available?

Reviewer #1: Yes

Reviewer #2: Yes

5. Is the manuscript presented in an intelligible fashion and written in standard English?

Reviewer #1: Yes

Reviewer #2: Yes

6. Review Comments to the Author

Reviewer #1: the authors have responded perfectly to the comments

Reviewer #2: The Authors have further improved the manuscript in this revised version.

This Reviewer is satisfied with the newly revised version.

7. PLOS authors have the option to publish the peer review history of their article (what does this mean?). If published, this will include your full peer review and any attached files.

Reviewer #1: No

Reviewer #2: **Yes: **Roberto Chiarle

---

## [Editor Report · Acceptance letter]

12 Sep 2022

PONE-D-22-16001R1 

Computational gene expression analysis reveals distinct molecular subgroups of T-cell prolymphocytic leukemia 

Dear Dr. Seifert:

I'm pleased to inform you that your manuscript has been deemed suitable for publication in PLOS ONE. Congratulations! Your manuscript is now with our production department. 

Kind regards, 

on behalf of

Dr. Francesco Bertolini 

Academic Editor

PLOS ONE